# Proteoform Patterns in Hepatocellular Carcinoma Tissues: Aspects of Oncomarkers

**DOI:** 10.3390/proteomes13030027

**Published:** 2025-07-01

**Authors:** Elena Zorina, Natalia Ronzhina, Olga Legina, Nikolai Klopov, Victor Zgoda, Stanislav Naryzhny

**Affiliations:** 1Institute of Biomedical Chemistry, Pogodinskaya, 10, Moscow 119121, Russia; el.zorina@ibmc.msk.ru (E.Z.); vic@ibmh.msk.su (V.Z.); 2Petersburg Nuclear Physics Institute Named by B. P. Konstantinov of National Research Centre “Kurchatov Institute”, Gatchina 188300, Russia; ronzhina@list.ru (N.R.); olgaleg@mail.ru (O.L.); klopov_nv@pnpi.nrcki.ru (N.K.)

**Keywords:** hepatocarcinoma, oncomarker, proteoforms, 2D electrophoresis

## Abstract

Background: Human proteins exist in numerous modifications—proteoforms—which are promising targets for biomarker studies. In this study, we aimed to generate comparative proteomics data, including proteoform patterns, from hepatocellular carcinoma (HCC) and nonmalignant liver tissues. Methods: To investigate protein profiles and proteoform patterns, we employed a panoramic, integrative top-down proteomics approach: two-dimensional gel electrophoresis (2DE) coupled with liquid chromatography–electrospray ionization–tandem mass spectrometry (LC-ESI-MS/MS). Results: We visualized over 2500 proteoform patterns per sample type, enabling the identification of distinct protein signatures and common patterns differentiating nonmalignant and malignant liver cells. Among these, 1270 protein patterns were uniformly observed across all samples. Additionally, 38 proteins—including pyruvate kinase PKM (KPYM), annexin A2 (ANXA2), and others—exhibited pronounced differences in proteoform patterns between nonmalignant and malignant tissues. Conclusions: Most proteoform patterns of the same protein were highly similar, with the dominant peak corresponding to theoretical (unmodified) protein parameters. However, certain proteins displayed altered proteoform patterns and additional proteoforms in cancer compared to controls. These proteins were prioritized for further characterization.

## 1. Introduction

Liver tumors rank as the second leading cause of cancer-related mortality worldwide, with hepatocellular carcinoma (HCC) representing the most prevalent form. HCC is frequently diagnosed at advanced stages, underscoring the critical need for early detection strategies. The five-year survival rate correlates strongly with the disease stage, dropping to ≤20% in late-stage cases [1]. Major risk factors for HCC include
Liver cirrhosis (viral hepatitis B/C-induced and alcoholic/non-alcoholic steatohepatitis) [2].Genetic disorders (alpha-1-antitrypsin deficiency, tyrosinemia, and hemochromatosis) [3,4].Toxic liver injury (e.g., steroid hormone drugs) [3].Fatty liver disease (alcoholic or non-alcoholic, with non-alcoholic fatty liver disease (NAFLD) driving a global rise in HCC incidence) [5,6,7].

Currently, in clinical practice, a tumor-specific marker, alpha-fetoprotein (AFP), is used for screening risk groups, disease progression, and survival prognosis. The alpha-fetoprotein test proposed in 1964 to detect serum fractions of embryonic alpha-globulins as a diagnostic marker of HCC is still used in clinical practice [8,9,10]. AFP remains a widely used biomarker due to its low cost, ease of measurement, and wide availability. It should be noted that its sensitivity and specificity is insufficient for both early diagnosis and widespread screening. In this regard, it is important to search for new, more specific and sensitive markers, including proteins whose content increases or decreases in the tumor. A feature of a significant part of human proteins is their existence in several or many modifications—proteoforms. For example, the above-mentioned AFP exists in several variants of glycosylated forms (proteoforms). And the tests measuring the fucosylated form of AFP (AFP-L3) used in clinical practice show greater sensitivity and specificity (according to some studies, up to 95%) [11]. Another marker, which is also present in 50–60% of patients with HCC, is des-gamma-carboxy-prothrombin (DGP or PIVKA-II). It is an abnormal (non-modified) prothrombin that is formed due to vitamin K deficiency or impaired metabolism in the liver. In HCC, liver cells lose the ability to carboxylate prothrombin, which leads to an increase in the blood level of its unmodified form [12]. The examples of AFP-L3 and DGP show that measuring the content of specific proteoforms provides better results [13]. At the same time, a combination of AFP, DGP, and AFP-L3 allows a sensitivity of 94% and a specificity of more than 97% [14].

Therefore, the proteoforms are the promising objects in biomarker studies [15,16,17,18]. Over the past decades, the use of a combination of two-dimensional electrophoresis (2DE) with panoramic mass spectrometry (integrative top-down proteomics) has become a promising approach to studying proteoforms, which effectively implements the capabilities of these methods [16,19,20,21]. As a result, for each of the proteins presented in the sample, due to preliminary fractionation, it is possible to obtain information on proteoform patterns in various conditions. The comparative analysis of such patterns in the norm and in HCC allows for reducing the labor costs of searching for specific proteoforms as potential tumor markers. Therefore, obtaining information on proteoforms and their profiles is especially relevant, given the numerous changes within the human proteome in cancer diseases [22].

In prior work, we applied the same approach in the model experiment of HCC—comparative analyses of the liver and the cell line HepG2 [23]. The current study expands these findings to clinical samples, revealing critical insights into HCC-specific proteoform signatures.

## 2. Materials and Methods

All reagents used were sourced from Sigma-Aldrich Corp. (St. Louis, MO, USA), unless another manufacturer is specified. The remaining reagents were obtained from the following companies: Thermo Fisher Scientific, Waltham, MA, USA: dithiothreitol (DTT), protease inhibitor cocktail; GE Healthcare (Pittsburgh, PA, USA): IPG DryStrip (gel strips), IPG-buffers, DryStrip-coating liquid, Coomassie Brilliant Blue (CBB) R350; Promega Corp., (Madison, WI, USA): Trypsin Gold Human [23,24].

Liver tissue samples from patients (*n* = 5) with a histologically confirmed diagnosis of hepatocellular carcinoma (stage T1b N0 M0) were provided by the First Department of Abdominal Surgery and Oncology of the B.V. Petrovsky Russian Scientific Center of Surgery. Two samples, namely tumor tissue and control liver tissue, were obtained from each patient in test tubes containing the RNAlater stabilizing solution (Thermo Fisher Scientific, USA). Tissue pieces (HCC and the non-tumor part of the liver) weighing ~1 g were obtained at the Blokhin Institute (Moscow, Russia). The sample collection protocol was approved by the Institute ethics rules. Protocol No. 01/14/21 of the Meeting of the Medical and Ethical Committee of the B.V. Petrovsky Russian Scientific Center of Surgery. Informed consent was signed by all patients and donors. The samples after surgery were completely immersed in the test tubes with RNAlater solution, where they soaked for 24 h at room temperature. Then, the samples were stored in this solution at −80 °C.

### 2.1. Extraction of Proteins

For extraction, the sample was placed in a ceramic mortar with liquid nitrogen. The frozen sample was crushed with a ceramic pestle, and the resulting powder was transferred to 1.5 mL test tubes (Eppendorf, Hamburg, Germany) at approximately 200 mg per test tube. To exclude the proteolytic and chemical degradation of proteins, further sample preparation was carried out on ice. Phosphate-buffered saline (PBS) with protease inhibitors (500 μL) was added to each tube, pipetted, and then centrifuged (2 min, 5000× *g*, 4 °C). After centrifugation, the supernatant was removed. The procedure was repeated, and the resulting pellet in each tube was dissolved in 600 μL of lysis buffer (7 M urea, 2 M thiourea, 4% CHAPS, 1% dithiothreitol (DTT), 2% ampholytes, pH 3–10, and a mixture of protease inhibitors), processed 6 times with a SONOPULS HD 2070 ultrasonic homogenizer (Bandelin Electronic, Berlin, Germany) in the following mode: 2 s pulse/2 s pause (6 times), and centrifuged for 5 min at 10,000× *g* at a temperature of 4 °C. The supernatant was collected, divided into 100 μL (1 mg) aliquots, and stored at −80 °C. Five pairs of samples (control and tumor) were processed in this manner.

### 2.2. Filter-Aided Sample Preparation (FASP) Method

The panoramic proteomic analysis of the obtained extracts was performed using filter processing and subsequent mass spectrometry (the so-called Filter-Aided Sample Preparation (FASP) method) [25]. Centrifuge concentrators (Microcon YM–30, Merck, Madison, WI, USA) were used for this purpose. Extracts containing the required amount of protein (300 μg) were placed in concentrators and sequentially treated with solutions (a) for the reduction in disulfide bonds (100 mM DTT in 100 mM Tris-HCl, pH 8.5), (b) for the alkylation of sulfhydryl groups (50 mM iodoacetamide, 8 M urea, 100 mM Tris-HCl, and pH 8.5), and (c) for hydrolysis with trypsin (Trypsin Gold, Promega, Madison, WI, USA). Each of the steps was accompanied by preliminary mixing on a Yellowline TTS 2 shaker (IKA, Königswinter, Germany) for 20–30 s incubation in a Comfort Thermomixer (Eppendorf, Enfield, CT, USA) and centrifugation at 9800× *g* for 15 min at 20 °C. After each step, the sequentially used solutions were removed by adding 200 μL of washing solution (8 M urea in 100 mM Tris-HCl, pH 8.5). Reduction was performed by incubation for 1 h at 56 °C. Alkylation was performed by incubation for 1 h in the dark at 20 °C. Before hydrolysis, the samples were washed twice with 200 μL of buffer solution (50 mM ammonium bicarbonate, pH 8.5). For hydrolysis, trypsin (Trypsin Gold, Promega, USA) in a buffer solution of 50 mM ammonium bicarbonate, pH 8.5, was used. The trypsin concentration required for hydrolysis was taken based on the mass ratio of the total mass of the enzyme/the total mass of the protein—1/100. Incubation was carried out overnight at 37 °C. To stop trypsinolysis, 50 μL of 30% formic acid solution was added to the hydrolysates on the filters. The resulting peptide solution was transferred to clean 250 μL glass inserts (Agilent, Santa Clara, CA, USA) and dried in a vacuum concentrator (Concentrator 5301, Eppendorf, Germany) at 45 °C. Before MS analysis, the dried peptides were dissolved in 20 μL of a 5% formic acid solution. The final concentration was 15 μg/μL, based on the calculations that the mass of the total protein taken for analysis is equal to the mass of peptides in the sample.

### 2.3. Two-Dimensional Gel Electrophoresis (2DE)

The procedure of 2DE was performed as before [26,27]. In short, samples (300 μg) (see Section 2.1) were mixed with rehydration buffer (7 M urea, 2 M thiourea, 2% CHAPS, 0.3% DTT, 2% IPG buffer, pH 3–10, and 0.001% bromophenol blue) in a final volume of 150 µL and loaded into 7 cm IPG gel strips (pH 3–11) (GE Healthcare, Chicago, IL, USA) by passive rehydration at 4 °C overnight. IEF was conducted at 20 °C (9000 volt-hours) on a HoeferTM IEF100 instrument (Thermo Fisher Scientific, USA). After IEF, the strips were incubated (2 times for 10 min each) in an equilibration solution (50 mM Tris-HCl (pH 6.8), 6 M urea, 2% SDS, and 30% glycerol) initially containing 1% DTT and then 5% iodoacetamide. The strips were sealed by hot 0.5% agarose on the top of a 14% polyacrylamide gel, and the denaturing second-direction electrophoresis was carried out at r.t. at a constant power of 3.5 W per gel using the Hoefer miniVE system (gel size 80 × 90 × 1 mm, “GE Healthcare”) [26,27]. Gels were stained with Coomassie R350, scanned using ImageScanner III (GE Healthcare), and analyzed using Image Master 2D Platinum 7.0 (GE Healthcare) and SameSpot (TotalLab, Gosforth, UK). This gel was cut into 96 sections with determined coordinates [28,29]. Each section (~0.7 cm^2^) was shredded and treated with trypsin. Tryptic peptides were eluted from the gel using extraction solution (5% (*v*/*v*) ACN, 5% (*v*/*v*) formic acid) and dried in SpeedVac.

### 2.4. ESI LC-MS/MS Analysis

ESI LC-MS/MS was conducted using an Orbitrap Q-Exactive mass spectrometer (“Thermo Scientific,” Waltham, MA, USA), as was published in our previous papers [26,27]. The data produced were treated with SearchGUI (v. 4.3.9) [30] and PeptideShaker platform (v.1.16.45) using the following parameters: enzyme—trypsin; maximum of missed cleavage sites—2; fixed modifications—carbaidomethylation of cysteine; variable modifications—oxidation of methionine, phosphorylation of serine, threonine, tryptophan, and acetylation of lysine; the precursor mass error—10 ppm; the product mass error—0.01 Da; minimal peptide size—8. Swiss-Prot (August 2022) was used as a protein sequence database [26,27]. Data validation was performed using FDR. The threshold setting for validation for each level (PSM, peptide, and protein) was ≤1%. Proteins were considered identified if at least 2 unique peptide sequences for it were found [31]. Protein abundance was estimated by the exponentially modified PAI (emPAI) value [32].

### 2.5. Analysis and Statistical Processing of Data

In the case of data obtained by the FASP method, calculations of the emPAI(%)—emPAI of the particular protein/isoform normalized to the sum of all emPAIs in the sample—were performed in Microsoft Excel 2010 using the following formula:(1)$$emPAI(\%)=\frac{emPAI}{\sum_{emPAI}}\times 100\%$$

For the sectional 2DE analysis, calculation of emPAI(%) was performed by normalizing the sum of emPAIs for the proteoforms of the individual protein to the sum of the emPAI values of all identified proteoforms in the sample.(2)$$emPAI(\%)=\frac{\sum_{emPAI}p_{1}+p_{2}+p_{3}+\ldots +p_{n}}{\sum_{emPAI}}\times 100\%$$
where $\sum_{emPAI}$ p_1_ + p_2_ + p_3_ + p_n_ is the sum of emPAI values of proteoforms related to a specific isoform of the protein-coding gene (PCG), and $\sum_{emPAI}$ is the sum of emPAI values of all identified proteoforms in the sample. The resulting normalized emPAI value reflects the absolute quantitative content of protein in the molar percentage concentration for a specific PCG in the sample.

A statistical assessment of the reliability of the results was performed using Student’s *t*-test with a significance level of *p* < 0.05. For further analysis, proteins corresponding to the following criteria were selected in each of the analyzed groups: fold change (FC) ≥ 1.5 and/or presence only in malignant cells. Visualization of the statistical data was performed in the Graph Pad Prism program (version 8.0.1) (https://www.graphpad.com accessed on 22 December 2024).

Functional annotations of the identified proteins in three categories, biological processes, molecular functions, and protein classes, were performed using the PANTHER web resource [33]. Graphical representation of the proteoform patterns was performed in the form of three-dimensional graphs in the Microsoft Excel program.

## 3. Results

### 3.1. Proteomic Profiling of Samples from Patients with HCC

#### 3.1.1. Panoramic Proteomic Profiling

The data on the identified proteins in all five samples were processed (Figure 1). Additionally, 3384 proteins were detected in tumors and 2824 in control samples. In total, 3677 non-redundant proteins were detected in all samples (Appendix A). Statistical processing allowed us to identify proteins whose levels significantly differed between the control and the tumor. It turned out that the level of 1627 proteins was higher in the tumor (FC cut off—1.5) or that they were not detected in the control. In the control, there were 656 of such proteins (Appendix A)

The graph of the relationship between the tumor and control liver tissue protein abundances (emPAI) shows that the balance of proteins with different copy numbers is maintained under different states (control or tumor) (Figure 2).

Statistically significant differences in the FC (fold change) between samples (T/C) can be observed by constructing a Volcano plot (Figure 3). Only 1613 proteins that were detected in both normal and tumor samples were considered in this case. Thus, according to statistical significance (*p* < 0.05), there are 85 differently expressed proteins (FC ≥ 1.5 up/down), of which 28 proteins had an increased abundance and 57 proteins had a decreased abundance in HCC (Figure 3). The most upregulated proteins (FC ≥ 4) are as follows: FBLN3 (EGF-containing fibulin-like extracellular matrix protein 1), G6PD (Glucose-6-phosphate 1-dehydrogenase), BAP31 (B-cell receptor-associated protein 31), ITAM (Integrin alpha-M), and HLAC (HLA class I histocompatibility antigen, C alpha chain). The most downregulated proteins (FC ≥ 3) are as follows: DHB8 ((3R)-3-hydroxyacyl-CoA dehydrogenase), AL8A1 (2-aminomuconic semialdehyde dehydrogenase), TKFC (triokinase/FMN cyclase), MMSA (methylmalonate-semialdehyde dehydrogenase), PAHX (phytanoyl-CoA dioxygenase, peroxisomal), ACOT2 (acyl-coenzyme A thioesterase 2, mitochondrial), GLYAT (glycine N-acyltransferase), DDAH1 (N(G), N(G)-dimethylarginine dimethylaminohydrolase 1), and F16P2 (Fructose-1,6-bisphosphatase isozyme 2).

#### 3.1.2. Two-Dimensional Sectional Proteomic Profiling

Building on our previous comparative proteomics analysis of normal liver tissue and HepG2 cell lines [23], we applied a sectional 2DE approach to extend panoramic proteomic profiling to HCC patient-derived tumor and control liver tissues. As an example, Figure 4 shows 2DE for tumor (T1) and control (C1) liver tissue from patient #1, further analysis of which allowed for the visualization of proteoform data as three-dimensional plots of proteoform patterns, which reflect the position of proteoforms in the gel and the relative protein content. As a result, a total of 13,000 proteoforms were detected for the T1 sample. Based on these data, three-dimensional graphs for proteoform patterns of each protein (isoform, more precisely) were constructed. Patterns of 2843 isoforms were constructed for the T1 sample, while 2427 were for the control sample (C1). Altogether, 4122 non-redundant proteoform patterns were found for all tumor samples and 3510 were found for all control samples.

The data obtained by the 2DE sectional profiling were collected and normalized according to Section 2.4. As in the case of proteomic profiling, a summary table was generated containing data on all identified proteins and their normalized emPAI values. Proteins that met the selection criteria (FC ≥ 1.5 or presence only in tumor tissue) were selected. A total of 4526 non-redundant proteins were identified in all samples of tumor and control liver tissue. Comparing the identified proteins in the tumor and the control liver tissue, 3106 proteins were found to be common. When comparing proteoform patterns obtained by 2DE sectional proteomic profiling, it turned out that in the overwhelming majority of cases, the patterns are very similar, and the dominant peak corresponds to the theoretical parameters of the protein (so it is unmodified) (Appendix A). Some examples are presented in Figure 5. There are also proteins with different patterns that are presented in Appendix A.

#### 3.1.3. Combining the Results of Panoramic and 2DE Sectional Proteomic Profiling

Using 2DE sectional proteomic profiling, we were able to identify significantly more proteins than with panoramic profiling in tumor tissue and control liver tissue (Figure 6).

For further analysis, the results presented as summary tables for panoramic and 2DE sectional profiling were combined into one table (Appendix A). The table shows the calculated data on the absolute quantitative protein content in the molar percentage concentration for a certain gene in tumor and control liver tissue obtained by two approaches. From the identified proteins, proteins with a cutoff criterion of FC ≥ 1.5 were selected for the more detailed analysis. Having analyzed the obtained data, not only on differently presented proteins but also on their proteoform patterns, we selected 38 proteins for which the changes in patterns were most pronounced compared to the control (Table 1, Figure 7, Appendix A).

There are three proteins in this list that are also among the statistically significantly up-regulated proteins detected by panoramic profiling (Appendix A): EGF-containing fibulin-like extracellular matrix protein 1 (FBLN3), Integrin alpha-M (ITAM), and very-long-chain enoyl-CoA reductase (TECR).

### 3.2. Functional Annotation of Potential HCC Biomarkers

Gene ontology analysis by Panther GO (https://www.pantherdb.org/ accessed on 13 December 2024) allowed us to compare proteins prevalent in the tumor or control by function, process, and class (Figure 8). Among the most characteristic differences, it is necessary to note the activation of immune processes (immune system process (GO:0002376)) and the process of interspecies interaction (biological process involved in interspecies interaction between organisms (GO:0044419)) in tumors. As for molecular functions, translation regulator activity (GO:0045182) and translational proteins (PC00263) prevail in HCC. The molecular function regulator activity (GO:0098772) is also more diverse in tumors. More proteins of the extracellular matrix (PC00102), DNA metabolism (PC00009), and cytoskeleton (PC00085) were detected as well. However, small molecule metabolism enzymes (metabolite interconversion enzymes, PC00262) are better represented in the control. But it is interesting that among the 38 selected proteins (Table 1), this class of enzymes is most prominent (seven gene entries—ACLY, PKM, AKR1B15, ACSL4, TECR, GALNT2, and ALPL).

In addition, we compared our data with the information from the Human Protein Atlas database (https://www.proteinatlas.org/ accessed on 22 March 2025) on prognostic markers of HCC, which in this database are mainly represented by transcription data. It turned out that 500 upregulated in HCC proteins appear as prognostic unfavorable markers in the Human Protein Atlas database. Interestingly, 70 proteins are classified as prognostic favorable markers. A slightly different picture is observed for proteins whose level prevails in the control (193 proteins). Here, there are approximately the same numbers of both favorable and unfavorable tumor markers (Appendix A). This indicates some ambiguity in determining the favorability of a protein based on determining the level of its RNA expression.

## 4. Discussion

Previously, we have published a paper where the search for HCC biomarkers was based on the comparative proteomics analysis of HepG2 cells and liver tissue [14]. Obviously, it was interesting to compare all available data (HepG2 cells, tumor tissue, and liver tissue). If we consider only the proteins for which an increase in their abundance (FC ≥ 1.5) in tumors and in HepG2 cells was detected, 747 common proteins were found out of 2094 (Appendix A). This information itself points out that HepG2 cells can be used for searching for HCC biomarkers. Also, 10 proteins from Table 1 (ACLY, ACSL4, ANXA2, ANXA3, COPD, EPIPL, TRFL, P3H1, KPYM, and SRSF5) were detected previously in the HepG2 cell line as having similar patterns to HCC proteoform (http://2de-pattern.pnpi.nrcki.ru/index.html accessed on 12 June 2025) [23]. This means that despite the considerable difference in growing conditions between HCC tissue and HepG2 cells that affect the cellular metabolism [34,35], still some proteomics peculiarities remain the same in both types of cells [36]. However, it is clear that the proteome of HepG2 cells in many aspects is different from the proteomes of the primary normal and tumor cells [37,38,39].

It should be mentioned that based on just their elevated abundance in HCC, there are many candidates for HCC biomarkers. The evident changes in proteoform patterns are quite rare cases, and the difference between the analyzed objects is mainly associated with changes in protein abundances. As the better (so-called ideal) HCC biomarkers are the main aim of our study, they should be searched for inside these altered patterns. An “ideal biomarker” should also have a high potential for secretion into the blood to be detected and quantified by non-invasive methods. According to UniProt, eleven proteins from Table 1 (ANXA1, ANXA2, FBLN3, FCN1, GALT2, TRFL, P3H1, PIGR, PROS, TSP1, and VWA1) are secreted and can be considered as better biomarker candidates for HCC. From this point of view, the proteins ANXA2, TRFL, and P3H1 look most interesting as their proteoform patterns are changed in HCC and HepG2 cells. The next step is to decipher the peaks (proteoforms) that are present in the tumor cells, not in the normal liver. This is a task for top-down mass spectrometry.

It should also be mentioned that there is a resolution challenge in our experiments. A single PTM, like acetylation or phosphorylation, can produce a shift of pI~0.05 [40]. However, in our sectional 2DE analysis, the pI range of the sections is much bigger (0.7–0.8). This means that we are missing a separation of many proteoforms as they can be located in the same sections. Accordingly, at higher resolutions, more proteoforms and proteoform pattern differences between HCC and control should be revealed. Actually, the resolution can be improved by running bigger gels and analyzing smaller gel sections, but this will dramatically increase the time and expenses of the experiment.

## 5. Conclusions

A combination of panoramic proteomic profiling with 2DE sectional proteomic profiling (integrative top-down proteomics) allows for more reliable and convenient representations of information, including diverse proteoforms coded by the same genes (proteoform patterns). These patterns for the majority of proteins are very similar in all types of the analyzed samples (HepG2, human liver, HCC tissue, and HCC control tissue), where dominant peaks correspond to the theoretical parameters of the protein (meaning it is unmodified). But, for some proteins, there are differences that could be used for the generation of specific HCC biomarkers.

## Figures and Tables

**Figure 1 proteomes-13-00027-f001:**
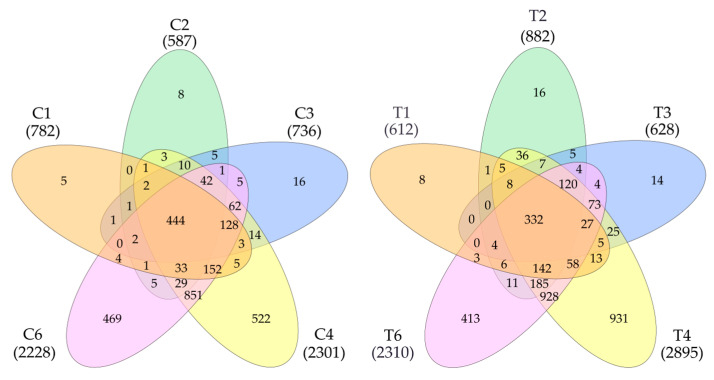
Venn diagram (https://www.interactivenn.net/ accessed on 18 December 2024) showing the distribution of detected proteins in control (**left**) and tumor samples (**right**).

**Figure 2 proteomes-13-00027-f002:**
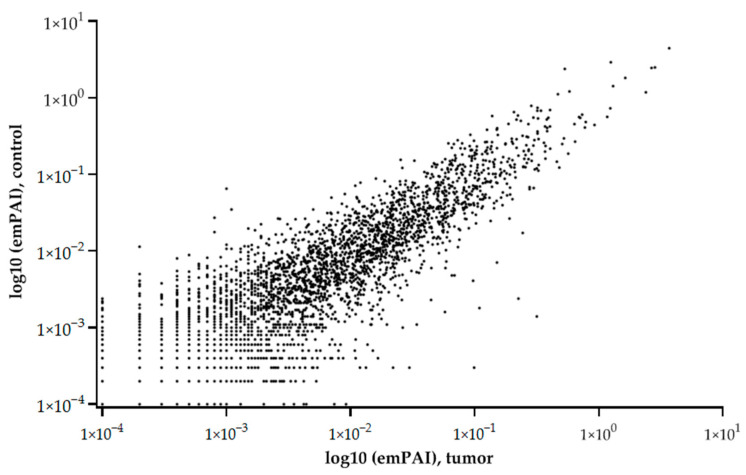
Relationship of average (5 samples) protein abundances (emPAI) between tumor and control liver tissue. The abscissa is the common logarithm of emPAI for the tumor, and the ordinate is the common logarithm of emPAI for the control liver tissue.

**Figure 3 proteomes-13-00027-f003:**
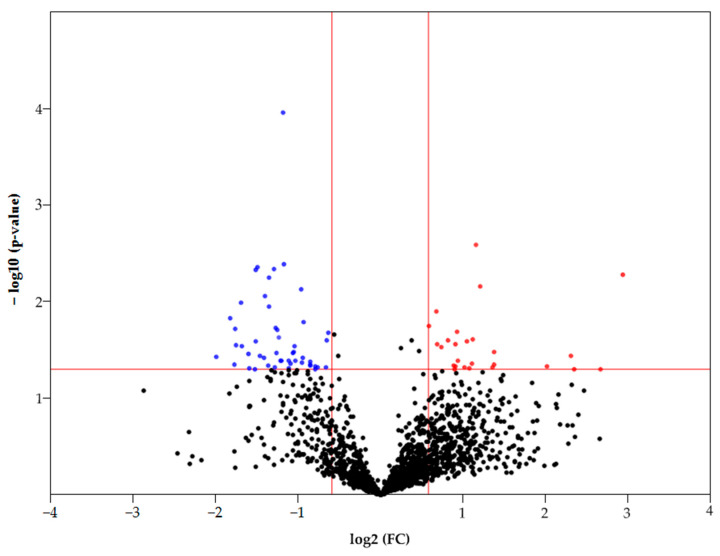
Volcano plot of FC of proteins that were detected in control and tumor samples. The colored dots represent proteins with *p* < 0.05 (see the red line crossing y-axis). Red dots represent proteins with increased levels in HCC (FC ≥ 1.5), blue dots represent proteins with increased levels in the control tissue (FC ≥ 1.5) (see the red lines crossing x-axis).

**Figure 4 proteomes-13-00027-f004:**
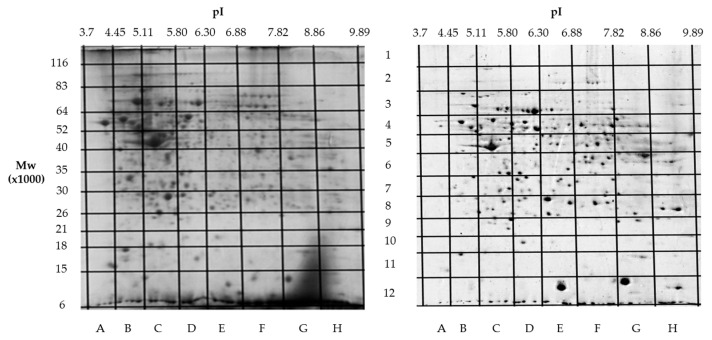
Two-dimensional electrophoresis (2DE) of samples from patient #1. The sections (marked with letters and numbers) with determined coordinates (pI/Mw) are shown. On the **left**—a tumor liver tissue (T1); on the **right**—a control liver tissue (C1).

**Figure 5 proteomes-13-00027-f005:**
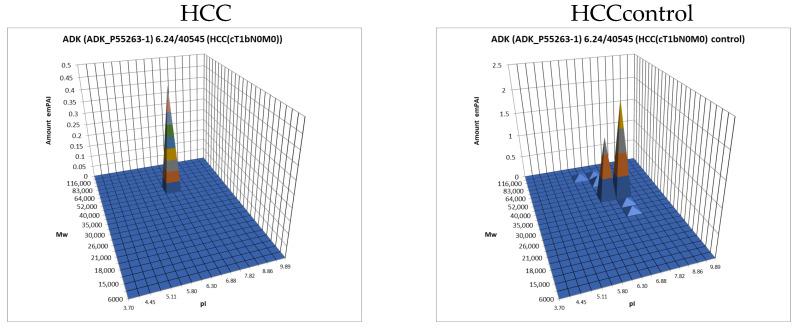
The examples of typical proteoform patterns similar for HCC and control HCC tissue.

**Figure 6 proteomes-13-00027-f006:**
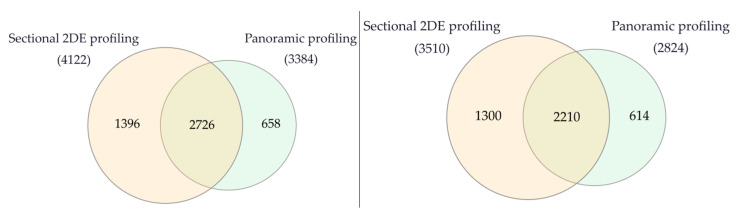
Venn diagram of proteins in liver tumor tissues (**left**) and control liver tissue (**right**) that were identified by 2DE sectional and panoramic proteomic profiling.

**Figure 7 proteomes-13-00027-f007:**
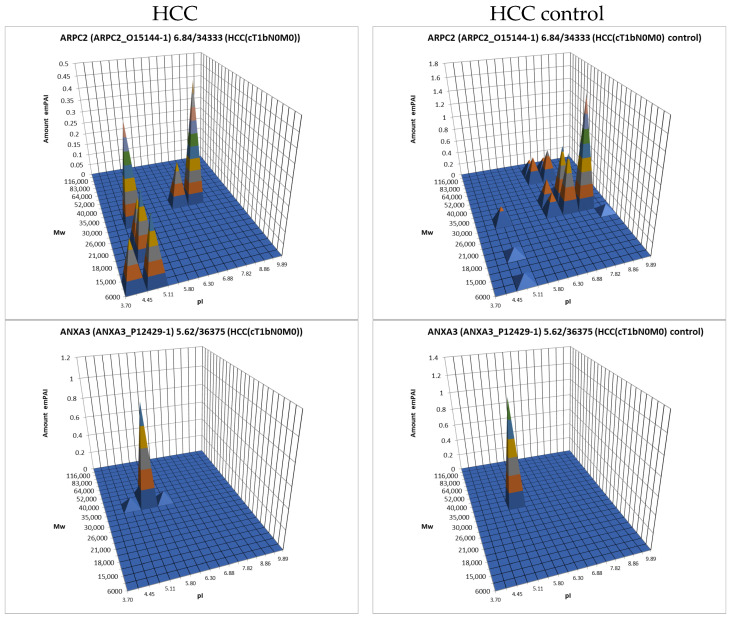
The graphical representation of proteoform patterns for some proteins from Table 1.

**Figure 8 proteomes-13-00027-f008:**
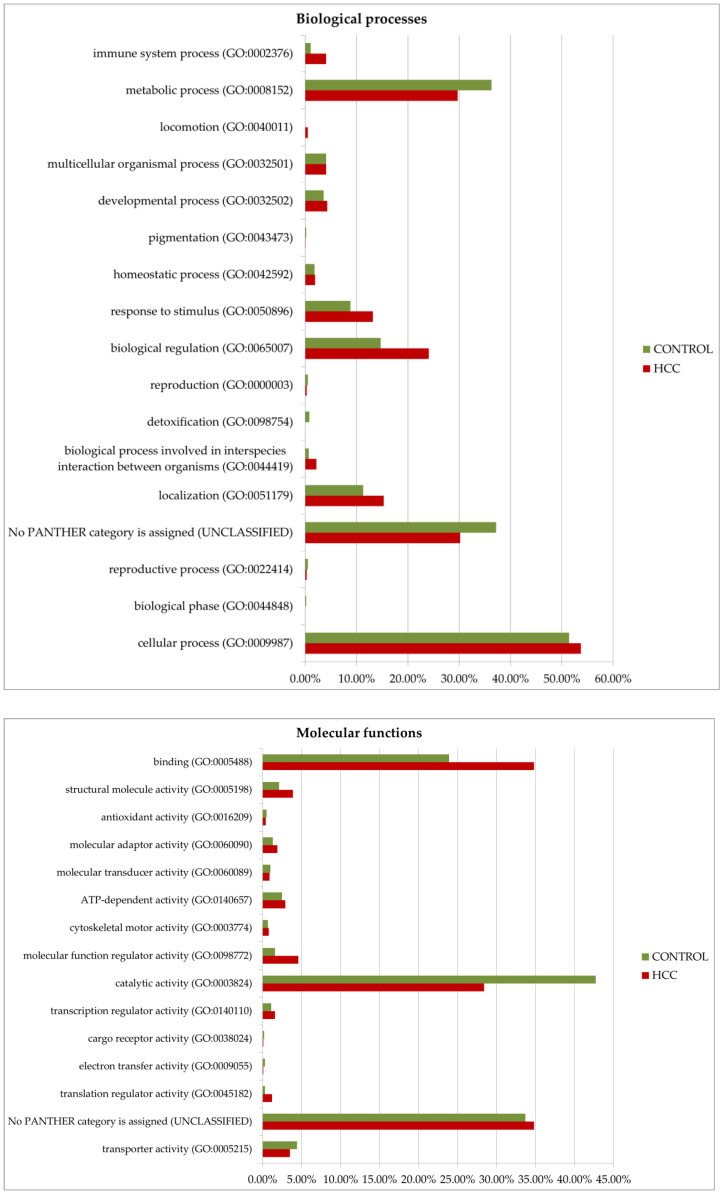
GO (gene ontology) analysis of proteins upregulated in HCC or in the control (FC ≥ 1.5). Protein distribution diagrams by biological processes, molecular functions, and protein classes are presented.

**Table 1 proteomes-13-00027-t001:** The selected proteins that are upregulated in HCC and have the most pronounced changes in proteoform profiles.

	Uniprot	Protein	Gene	Name	FC (2DE)	FC (FASP)
1	P53396	ACLY	*ACLY*	ATP-citrate synthase	17.24	1.79
2	O60488	ACSL4	*ACSL4*	Long-chain fatty acid–CoA ligase 4	375.38	N/A
3	C9JRZ8	AK1BF	*AKR1B15*	Aldo-keto reductase family 1 member B15	81.9	N/A
4	P05186	PPBT	*ALPL*	Alkaline phosphatase, tissue-nonspecific isozyme	12.8	N/A
5	P04083	ANXA1	*ANXA1*	Annexin A1	1.62	2.19
6	P07355	ANXA2	*ANXA2*	Annexin A2	1.85	1.58
7	P12429	ANXA3	*ANXA3*	Annexin A3	7.5	6.32
8	P48444	COPD	*ARCN1*	Coatomer subunit delta	3.2	1.99
9	P40121	CAPG	*CAPG*	Macrophage-capping protein	31.22	N/A
10	P02741	CRP	*CRP*	C-reactive protein	35.27	N/A
11	P08311	CATG	*CTSG*	Cathepsin G	14.28	2.68
12	Q12805	FBLN3	*EFEMP1*	EGF-containing fibulin-like extracellular matrix protein 1	11.39	7.68
13	P58107	EPIPL	*EPPK1*	Epiplakin	29.61	N/A
14	O00602	FCN1	*FCN1*	Ficolin-1	11.97	N/A
15	Q10471	GALT2	*GALNT2*	Polypeptide N-acetylgalactosaminyltransferase 2	32.8	3.97
16	P04062	GLCM	*GBA*	Lysosomal acid glucosylceramidase	23.83	N/A
17	P30511	HLAF	*HLA-F*	HLA class I histocompatibility antigen, alpha chain F	8.09	N/A
18	P11215	ITAM	*ITGAM*	Integrin alpha-M	N/A	4.96
19	P05107	ITB2	*ITGB2*	Integrin beta-2	14.89	2.39
20	P80188	NGAL	*LCN2*	Neutrophil gelatinase-associated lipocalin	13.56	2.95
21	P17931	LEG3	*LGALS3*	Galectin-3	2.21	N/A
22	P02788	TRFL	*LTF*	Lactotransferrin	21.11	3.74
23	O00187	MASP2	*MASP2*	Mannan-binding lectin serine protease 2	2.51	N/A
24	P23368	MAOM	*ME2*	NAD-dependent malic enzyme, mitochondrial	5.58	1.56
25	P41218	MNDA	*MNDA*	Myeloid cell nuclear differentiation antigen	41.75	N/A
26	P05164	PERM	*MPO*	Myeloperoxidase	19.12	3.21
27	Q32P28	P3H1	*P3H1*	Prolyl 3-hydroxylase 1	2.14	N/A
28	P01833	PIGR	*PIGR*	Polymeric immunoglobulin receptor	104.61	N/A
29	P14618	KPYM	*PKM*	Pyruvate kinase PKM	3.38	2.76
30	P07225	PROS	*PROS1*	Vitamin K-dependent protein S	9.53	1.86
31	Q96M27	PRRC1	*PRRC1*	Protein PRRC1	4.19	1.64
32	P08575	PTPRC	*PTPRC*	Receptor-type tyrosine-protein phosphatase C	14.94	2.64
33	Q8NC51	PAIRB	*SERBP1*	Plasminogen activator inhibitor 1 RNA-binding protein	18.9	N/A
34	Q13243	SRSF5	*SRSF5*	Serine/arginine-rich splicing factor 5	2.92	N/A
35	Q9NZ01	TECR	*TECR*	Very-long-chain enoyl-CoA reductase	1.94	1.92
36	P02786	TFR1	*TFRC*	Transferrin receptor protein 1	77.86	2.75
37	P07996	TSP1	*THBS1*	Thrombospondin-1	11.11	3.42
38	Q6PCB0	VWA1	*VWA1*	von Willebrand factor A domain-containing protein 1	39.24	N/A

## Data Availability

The data that support the findings of this study are openly available in http://2de-pattern.pnpi.nrcki.ru/index.html (accessed on 20 June 2025).

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
