# Peer review of "Proteoform Patterns in Hepatocellular Carcinoma Tissues: Aspects of Oncomarkers"

_proteomes, 2025, doi:10.3390/proteomes13030027_

Round 1

Reviewer 1 Report

Comments and Suggestions for Authors

Overall this study needs to be improved but is appropriate for publication after revisions are made. 

Fundamentally there are issues with some details of how the study was performed and which/how many samples were used, and which samples the data are related to. 

The supplementary data is not accessible which has made verifying the conclusions impossible and this cannot be published without thorough review of that data. 

Also I note that the title mentions top down proteomics however this manuscript does not show any top down data. this should be removed for clarity. 

Likewise sequence information, total protein ID's, patient information etc is all also missing. i am assuming that this is in the supplemental data however as stated, this is not accessible. 

Detailed in line comments are provided in the attached PDF. These must be addressed prior to publication. 

I look forward to receiving the revised manuscript 

Reviewer 2 Report

Comments and Suggestions for Authors

The manuscript titled “Proteoforms patterns in the samples of hepatocellular carcinoma: aspects of oncomarkers” shows an interesting proteomic investigation aiming to discover hepatocellular carcinoma biomarkers profiling protein proteoforms. Although the manuscript is clearly organized,  both the experimental part and the description of results omit relevant information and clear descriptions of the workflow applied to data elaboration which need a major revision. The authors described in the abstract that a top-down approach was applied to proteomic analysis, however the proteomic results presented are from protein digests. This is not clear. Following detailed comments are listed:

Materials and Methods section:

-The details of several chemicals used for the experiments are omitted, i.e. PBS, protease inhibitors, Urea, CHAPS , organic solvent, as few examples. Specifications of the software used for Venn diagram elaboration is omitted.

-The number of patients enrolled in the investigation (n=5) should be reported in the first part when collection of samples is described.

Paragraph 2.2: A total protein content of 300 ug was loaded in FASP device, is quite high content, what were the specifications of these devices regarding the efficiency of protein recovery based on the loaded quantity? The final volume of elution of the total protein loaded in FASP was reported as 20 uL of 5% formic acid solution. This resulted in a final theoretical concentration of 15 ug/uL, whereas the authors reported a final concentration for LC-MS analysis of 1 ug/uL instead. Therefore, were the samples further diluted before LC-MS analyses? If so, it has to be specified.

- Paragraph 2.3: the description of the experimental procedure is not clear: the samples prepared according to section 2.1 are finally corresponding to different aliquots of surnatant, each of 100 uL. Therefore, each aliquot was added of 50 uL of rehydration buffer, since the final volume was 150 uL? Did it happen this way or a different way? This is not clear.

- Paragraph 2.4: acetylation PTM can occur on different amino acid residues, while only lysine was selected in the study? Furthermore the version of protein sequence database used as reference for protein identification is quite dated (October 2014).

-The authors reported that “only 100% confident results were selected”, but, what does 100% refer to? What level of confidence does 100% refer to? High, medium ? How were identification data validated, based on FDR? What were the parameters set? Several settings of protein identification data elaboration were omitted, as, i.e., the minimum peptide length …

- protein identification was set with characterization of at least 2 unique proteolytic peptides. If 2 is the minimum number of peptides necessary for identification of a protein, the method of abundance calculation taking into account the number of characterized relative proteolytic fragments could be inconsistent. A truncated proteoform generates a lower number of proteolytic fragments but its content could be either consistent as other proteoforms or the intact parent protein of reference. This is not clear and should be better explained.

Results section:

Paragraph 3.1.1: Several figures are difficult to read, the font of graphs’ axis text or the numbers in the figures  are often too small to be easily read (see Figures 1, 5, 7).

The workflow applied to the results data and to data elaboration is not clear, below detailed comments are reported:

-it is not clear how many proteins have been identified in CTRL samples, 3198-2824, thus 374 proteins? In the Venn diagram of Figure 1 even just the number of proteins common to all the 5 samples analysed is >374. Totally the number should be 3572 proteins, while total number of identified proteins reported in text is 3677. What does this number refer to? It is not clear where the number of 1627 higher in tumour proteins originates, which should have been identified both in pathological and control samples, precisely because they are higher and not exclusive. Which proteins does the graph in figure 2 consider? Further down in the text, 1613 proteins detected in both normal and tumour samples are reported, this is not clear, because this number should be actually the 1627 before reported. Then authors are talking about 85 differently expressed proteins, which should be part of these 1613 or 1627, but this is confusing. It is very difficult to follow the data processing steps.

Paragraph 3.1.2:

-The numbers in figure 6 do not match the other numbers reported throughout the text before.

-In table 1 the gene name should be reported and possibly in addition the name of the protein, the third column of the tables can be omitted. In table 1 some gene names are in bold, why? This should be discussed in the text and reported in a footnote of the table.

-Paragraph 3.2: Which group of proteins was analysed for bioinformatic elaboration? This is not specified. Table 2 contains a partial list of protein, the table should be eliminated and reported as supplementary material with the complete list of proteins.

-Figure 5 and 7: data from a previous study seem reported here (HepG2 cell and normal liver tissue), this should be specified by referring to the relative bibliographic reference, or, if they are new data from this study, this is not reported in the materials and methods. This is also valid for the paragraphs of the -Discussion and Conclusions also evaluating these previous data.

-“ANXA2, TRFL and P3H1 look most interesting as their proteoform patterns are changed…” authors talk about proteoforms but they have not been identified, this will be a task of top down proteomics, therefore top-down proteomics has not been applied here as the analytical approach reported in the abstract.

As general comment, the comparison of tumour samples and controls should concern the proteins identified in all samples of each type, and therefore considering only the proteins common to all sample type, which is the shared number of protein of the relative Venn diagrams elaboration. Since few samples have been analysed, analysing and comparing only the proteins identified with repeatability in pathological or control samples, avoiding the data variable in the individual samples,  strengthens the potential to identify possible biomarkers. Proteoforms are cited along all manuscript however they were not characterized, only graphical representations are presented, what proteoforms have been identified, with what confidence? what PTMs were identified following the parameters applied in the search?

It is advisable to specify in the title that samples are tissues; therefore “in the sample of hepatocellular carcinoma” should be replaced by “in hepatocellular carcinoma tissues”

Round 2

Reviewer 1 Report

Comments and Suggestions for Authors

All comments have either been answered or addressed. the manuscript should be published in its current form. 

Author Response

Comment  (All comments have either been answered or addressed. the manuscript should be published in its current form.)  Response. Thank you! 

Reviewer 2 Report

Comments and Suggestions for Authors

The manuscript “Proteoform patterns in hepatocellular carcinoma tissues: aspects of oncomarkers” has been revised according to the suggested alterations.

Minor comments:

-The specifications of some chemicals are still missing, i.e., Urea, Thiourea, CHAPS, PBS, iodoacetamide, Tris-HCl

- the text “-“ at line 196 before the number 2824 of the proteins identified in HCC Control should be removed.

-Supplementary Table S1: the sheet titled “HCC FASP Control” does not include data, it is a blank page.

Author Response

Comment 1 (The specifications of some chemicals are still missing, i.e., Urea, Thiourea, CHAPS, PBS, iodoacetamide, Tris-HCl). Response (See In the text - "All reagents used were sourced from Sigma-Aldrich Corp. (St. Louis, MO, USA), unless another manufacturer is specified". 

 Comment 2 ( “-“ at line 196 before the number 2824 of the proteins identified in HCC Control should be removed).   Response. (Agree. The correction was done -  "3384 proteins were detected in tumors and 2824 in control samples. Totally, 3677 non-redundant proteins were detected in all samples".

Comment 3 (-Supplementary Table S1: the sheet titled “HCC FASP Control” does not include data, it is a blank page.)  Response. (Agree, sorry, it's our fault. Corrected).